# Microbiota regulates the turnover kinetics of gut macrophages in health and inflammation

Qi Chen, Sajith Nair, Christiane Ruedl

The gut immune system has evolved to co-exist in a mutually beneficial symbiotic relationship with its microflora. Here, using a germ-free fate-mapping mouse model, we provide clear insight into how the enteric commensals determine the kinetics of macrophage turnover. The microbiome density along the gastrointestinal tract defines the persistence of ontogenically diverse macrophages, with the highest numbers of the long-lived F4/80$^{hi}$Tim4$^+$ macrophage subset in the less densely colonized small intestine. Furthermore, the microbiome contributes to a tightly regulated monocyte-dependent replenishment of both long- and short-lived F4/80$^{hi}$ macrophages under homeostatic and inflammatory conditions. In the latter situation, the commensals regulate rapid replenishment of the depleted macrophage niche caused by the intestinal inflammation. The microbial ecosystem imprints a favorable cytokine microenvironment in the intestine to support macrophage survival and monocyte-dependent replenishment. Therefore, the host immune system-commensal cross-talk provides an efficient strategy to assure intestinal homeostasis.

## Introduction

The microbiota and its metabolites not only play a crucial role in inducing and shaping the host immune system, but have also been implicated in the modulation of several infectious, inflammatory, and metabolic diseases (Arrieta et al, 2015; Knip & Siljander, 2016; Forbes et al, 2018; Meijnikman et al, 2018). Remarkably, both host and commensals have evolved to communicate and maintain a beneficial symbiotic relationship, in particular in the gut, where the most commensals are residing.

Among the intestinal immune cells, macrophages are crucial in controlling gut homeostasis, wound healing and epithelial repair (Joeris et al, 2017; Bain & Schridde, 2018; Na et al, 2019; Wang et al, 2020; Viola & Boeckxstaens, 2020a; Nobs & Kopf, 2021). In addition, they maintain the delicate balance between sensing invading microorganisms and tolerance toward harmless enteric commensals and dietary antigens. Furthermore, under certain circumstances,

macrophages promote inflammatory bowel disease (Grainger et al, 2017) and carcinogenesis (Soncin et al, 2018; Mola et al, 2020). Therefore, improved understanding of the cross-talk between macrophages and the gut microflora have potentially important therapeutic implications.

Most intestinal resident macrophages are strategically positioned in the lamina propria (LP) of the mucosa to protect against invading microorganisms and respond to food antigens. Therefore, LP macrophages, together with intestinal DCs, are the main players involved in balancing microbial defense and oral tolerance (Bain & Schridde, 2018). In addition, a tiny population of gut-resident macrophages is located strategically in the area of the muscularis externa, where their proximity to sympathetic neurons of the enteric nervous system influences gut motility (Muller et al, 2014; De Schepper et al, 2018; Viola & Boeckxstaens, 2020b).

In general, tissue-resident macrophages originate from yolk sac or fetal liver embryonic precursors and self-renew in situ to maintain their numbers throughout life (Sheng et al, 2015; Ginhoux & Guilliams, 2016; Perdiguero & Geissmann, 2016). In contrast, most of the fetal-derived gut macrophages residing in the LP are quickly replaced after birth by monocyte-derived macrophages, and the cell pool depends on constant replenishment by circulating monocytes. This process occurs not only during inflammation, but also under steady-state conditions (Smythies et al, 2006; Bain et al, 2013, 2014; Bain & Mowat, 2014). A small fraction of the gut macrophage population located in the muscularis externa is long-lived and the numbers are maintained independently through BM input as a result of self-renewal (De Schepper et al, 2018). Therefore, the intestine represents a unique site in which distinct macrophage subpopulations co-exist with different origins and exhibit replenishment kinetics (Bleriot et al, 2020).

LP-resident macrophages are located in the proximity of the intestinal epithelium, which separates mucosal immune cells from the intestinal lumen with its myriad of endogenous microbiota; therefore, it has been speculated that monocyte-replenishment of gut LP macrophages is mediated by local low-grade inflammation caused by the enteric commensals. Furthermore, initial evidence showed that the macrophage turnover could be reduced by antibiotic treatment (Bain et al, 2014) and decreasing numbers of intestinal macrophages were detected in mice maintained under

---

Nanyang Technological University, School of Biological Sciences, Singapore

Correspondence: Ruedl@ntu.edu.sg

germ-free (GF) conditions compared to their counterparts maintained under specific pathogen-free (SPF) conditions (Bain et al, 2014; Shaw et al, 2018; Kang et al, 2020). However, despite these recent advances, which have been obtained mainly in antibiotic-treated animals, the exact contribution of commensal bacteria to the kinetics of macrophage turnover under GF conditions remains to be elucidated.

To clarify the contribution of gut commensals in "driving" the steady-state turnover and life-span of LP-resident macrophages, we generated a GF fate-mapping *Kit*^MerCreMer^/*R26*^YFP^ transgenic mouse model that helped us circumvent some caveats of incomplete endogenous microflora depletion through antibiotic treatment and potential off-target drug effects (Kennedy et al, 2018). We compared the turnover kinetics of large and small intestine macrophages in the absence or presence of enteric microflora under steady-state and inflammatory conditions.

# Results

### Profile of intestinal myeloid cells in the presence or absence of microbiota

In a previous study, we identified a significant F4/80^hi^ resident macrophage subset and a Ly6C⁻MHCII⁺ macrophage subpopulation in the murine large intestine LP. The latter population was found to be derived directly from monocytes via a pathway known as the "monocyte waterfall" (Soncin et al, 2018). Here, in our multiparameter flow cytometry and uniform manifold approximation and projection (UMAP) analyses, we included an additional marker, the scavenger receptor Tim-4 (also known as Timd4), to delineate a long-lived resident macrophage subpopulation present in different tissues of the gastrointestinal tract (Scott et al, 2016; Shaw et al, 2018; Dick et al, 2019; Chen & Ruedl, 2020; Liu et al, 2020). In addition, to complete the myeloid cell profiling in the small and large intestines of SPF and GF mice, we included markers such as MHC II, CD11c, CD11b, CD103, Ly6C, Siglec-F, and Ly6G for analysis of DCs, monocytes, monocyte-derived macrophages, eosinophils, and neutrophils, respectively.

In both the small and large intestines, Siglec-F⁺ eosinophils were the main myeloid cell population, with higher numbers in GF mice than in SPF mice (Figs 1A–D and S1A–C), which was consistent with previous reports (Jimenez-Saiz et al, 2020). In contrast, the numbers of neutrophils, Ly6C^hi^ monocytes and monocyte-derived macrophages (Ly6C⁺MHCII⁺ and Ly6C⁻MHCII⁺) were significantly reduced in both GF colon and small intestines compared with their SPF counterparts (Figs 1A–D and S1A–C). There were no significant differences in the colonic CD11c^hi^MHCII^hi^ DC subpopulations (CD103⁺CD11b⁻, CD103⁺CD11b⁺, and CD103⁻CD11b⁺) of SPF and GF mice, whereas the numbers of CD103⁺CD11b⁻ and CD103⁻CD11b⁺ DC in the small intestine were significantly reduced in the GF intestinal microenvironment (Figs 1A–D and S1A–C).

Concerning resident gut F4/80^hi^ macrophages, commensals distinctly regulated the frequency and numbers of cells in the Tim-4⁻ and Tim-4⁺ subpopulations. In the SPF colon, with the densest microbial ecosystem, Tim-4⁻ macrophages were the predominant

F4/80^hi^ cell fraction (>80%). On the other hand, in the small intestine, where the density of commensal is 10–100 times lower, their long-lived Tim-4⁺ counterparts were the most numerous (Fig 1E). Accordingly, in the absence of the microbiota, long-lived Tim-4⁺ macrophages comprised the largest proportion of the resident macrophage population in both the colon and small intestine (Fig 1E). This difference was also reflected in their absolute numbers. Colonic and small intestine F4/80^hi^ Tim-4⁺ cell numbers were higher in GF mice, whereas the F4/80^hi^Tim-4⁻ cell numbers were augmented in SPF mice (Figs 1C and S1C).

In summary, the myeloid cell landscape is modulated by the endogenous microflora. Enteric eosinophils adapt to the GF microenvironment with increased absolute numbers, whereas other myeloid cells, such as neutrophils, monocytes, monocyte-derived macrophages, and F4/80^hi^ macrophages, require microbial colonization for recruitment and/or persistence in the intestine.

### The turnover kinetics of colonic resident F4/80^hi^ macrophages, but not of other myeloid cells, is controlled by the microbiota

Adult fate-mapping analysis was performed using SPF and GF *Kit*^MerCreMer^/*R26*^YFP^ mice, which allowed us to follow the population turnover kinetics of the colon LP myeloid cell population driven by the BM input in the absence and presence of commensals.

The labeling index of colonic tissue-resident macrophages (F4/80^hi^), monocytes (Ly6C^hi^MHCII⁻), monocyte-derived macrophages (Ly6C⁺MHCII⁺and Ly6C⁻MHCII⁺), DCs (CD11c^hi^MHCII⁺), eosinophils (Siglec-F⁺), and neutrophils (Ly6G⁺) was analyzed in adult mice at 2 mo after tamoxifen (TAM) administration (Fig 2A).

With the exception of resident macrophages, the replacement kinetics for all colonic myeloid cells, including monocytes, monocyte-derived macrophages neutrophils, DCs, and eosinophils, was comparable in SPF and GF mice (Fig 2B). Only about 25% of the colonic F4/80^hi^ macrophage cells in GF mice were YFP⁺ compared to 55% of the corresponding cells in the large intestines of SPF mice, indicating that the microbiota enhanced the turnover kinetics of resident F4/80^hi^ macrophages. In GF mice, most F4/80^hi^ macrophages had clearly maintained their embryonic origin and had not been replaced by BM-derived cells within the 2-mo labeling period. As shown in Fig 2C, this pattern of reduced YFP labeling was retained when F4/80^hi^ resident macrophages were subdivided into long-lived F4/80^hi^Tim4⁺ and short-lived F4/80^hi^Tim4⁻ resident macrophages. Both GF resident macrophage subsets were only slowly replaced by BM-derived cells and retained their fetal seed population for a more extended period than their SPF counterparts (Fig 2C).

Because the microbiota and their metabolites have location-specific signatures and densities (Donaldson et al, 2016), we divided distal and proximal colon sections for separate analysis. YFP-labeling indexes of F4/80^hi^Tim4⁺ and F4/80^hi^Tim4⁻ resident macrophages and Ly6C⁻MHCII⁺ monocyte-derived macrophages obtained from SPF and GF mice were analyzed at 4 mo post-TAM injection. In the distal colon section, colonized by the highest density of residing commensals (Donaldson et al, 2016), the YFP labeling of both GF F4/80^hi^Tim4⁺ and F4/80^hi^Tim4⁻ resident macrophage subpopulations were significantly reduced when compared to the SPF counterparts (Fig 2D). In the proximal colon tract,

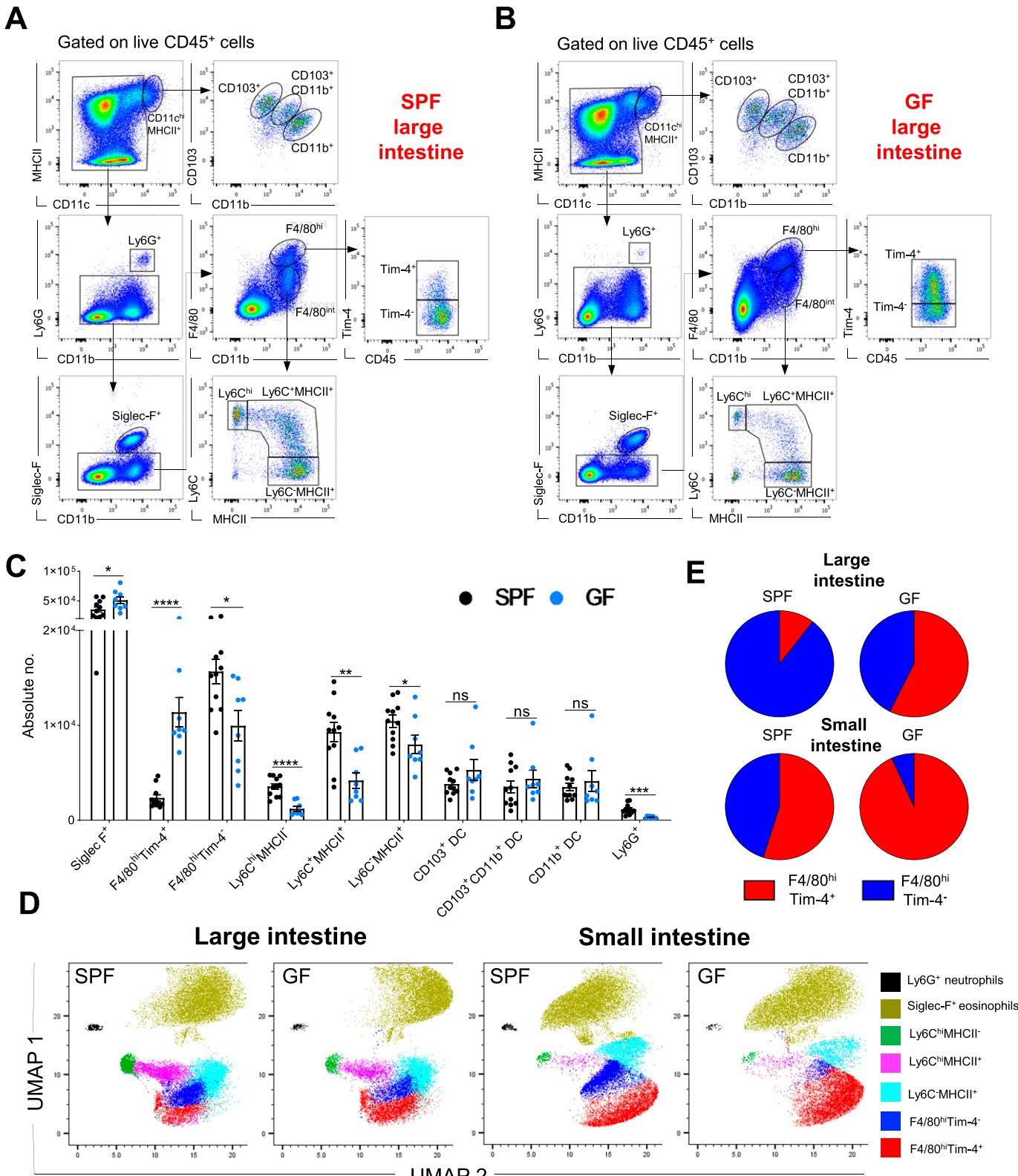

**Figure 1. Myeloid cell profile in the colon lamina propria of specific pathogen-free (SPF) and germ-free (GF) mice.**
**(A, B)** Representative hierarchical gating strategy used for flow cytometric analysis showing the following distinct colonic myeloid cell populations in SPF (A) and GF (B) mice: two different F4/80hi resident macrophages (F4/80hiTim-4+ and F4/80hiTim4−), three distinct DC subsets (CD103+CD11b−, CD103+CD11b+ and CD103−CD11b+), neutrophils (Ly6G+), monocytes (Ly6Chi), eosinophils (Siglec-F+), and monocyte-derived macrophages (Ly6C+MHCII+ and Ly6C−MHCII+). **(C)** Bar charts with individual data

with a less dense microbiota population than the distal tract, replenishment of the long-lived F4/80[hi]Tim4[+], but not the F4/80[hi]Tim4[−] macrophage subset in GF mice was significantly reduced compared to their SPF counterparts (Fig 2D). In the case of Ly6C[−]MHCII[+] monocyte-derived macrophages (Fig 2D) and all other myeloid cell populations (not shown), the YFP labeling was comparable in the distal and proximal colon sections of GF and SPF mice.

In summary, the microbiota dictates the replenishment kinetics of the LP-resident F4/80[hi] Tim4[+] and Tim4[−] macrophage populations only. In contrast, all other analyzed myeloid cells were swiftly replaced by BM-derived cells via a mechanism that is independent of the presence or absence of commensals in the colon.

## The replacement kinetics of small intestine resident F4/80[hi] macrophages is slower than the colonic counterparts and moderately affected by the microbiota

Because the bacterial density differs along the intestine (Tropini et al, 2017), we compared the replacement kinetics of myeloid cells in the proximal tract of the small intestine, which is known to harbor fewer microorganisms, with that in the large intestine (Kastl et al, 2020).

Like the large intestine, with the exception of resident LP F4/80[hi] macrophages, all the analyzed myeloid cells, including monocytes, monocyte-derived macrophages, eosinophils, and DCs, were fully replenished to an equivalent extent by BM cells in a manner that was independent of the presence of the local intestinal commensals. In contrast, the turnover kinetics of the resident LP small intestine F4/80[hi] population was retarded, with a YFP-positive labeling profile detected in only about 20% of these cells (Fig 3A and B). This rate of replenishment was even slower than that of their colonic counterparts, with 60% YFP-positive labeling detected at the corresponding time point (Fig 2B). In the absence of the microbiota, an additional, but slight decrease in YFP labeling was observed, although this did not reach the level of statistical significance (Fig 3A and B). Only when the LP F4/80[hi] macrophages were subdivided based on Tim-4 expression, a significant difference in the turnover of short-lived F4/80[hi]Tim4[−] macrophages between GF versus SPF mice was observed. In contrast, there was no difference in the turnover of the longer lived F4/80[hi]Tim4[+] macrophages between GF and SPF mice, probably because of their already very slow replenishment kinetics (~10% YFP labeling) (Fig 3C).

In summary, our fate-mapping analysis provided further evidence to confirm that gut LP-resident macrophages are replenished by BM-derived cells although at a slower rate than the other myeloid cells, which are swiftly replaced in both the small and large intestine via a mechanism that is independent of the local commensals. However, colon F4/80[hi] macrophages, residing in a microbe-rich environment, showed more rapid replenishment kinetics than

their counterparts in the small intestine counterparts, which were further be reduced in GF mice. These observations indicate that the density of microorganisms influence the resident macrophage turnover rate.

## The microbiota affects chemokine and cytokine expression in the intestine

The recruitment of monocytes into the gastrointestinal tract under steady-state and inflammatory conditions is regulated by chemokine–chemokine receptor interactions (Charo & Ransohoff, 2006); therefore, we analyzed the expression of macrophage/monocyte relevant chemokines in GF and SPF colons and small intestines. Notably, the expression of chemokine genes such as *Ccl2*, *Ccl5*, and *Ccl7* was significantly diminished in both large and small intestines of GF mice when compared to the gastrointestinal tract of SPF mice (Fig 4). Also, expression of *Csf1*, which encodes a growth factor known to support monocyte differentiation and macrophage survival, and the expression of inflammatory cytokine genes such as *Tnfα* and *Il1β*, was significantly diminished in both GF colon and small intestine. However, the local expression of other growth factors genes, such as *Csf2*, *Csf3*, and *Tgfβ1*, was not affected by the endogenous commensals (Fig 4). These data underline the capacity of enteric microbiota to modulate the expression of cytokines and chemokines in the intestinal microenvironment to support macrophage survival and monocyte recruitment.

## The microbiota supports monocyte repopulation of an empty macrophage niche during inflammation

Next, we examined the influence of gut commensals on repopulation of an empty macrophage niche under inflammatory conditions. It is well established that inflammation and infections lead to transient loss of resident macrophages resulting in an empty macrophage niche in many distinct organs/tissues (Barth et al, 1995; Ginhoux et al, 2017; Lai et al, 2018). We used a dextran sulfate sodium (DSS)-induced colitis mouse model to achieve this in the large intestine (Soncin et al, 2018). After administration of DSS to mice in their drinking water, flow cytometric analysis confirmed that colonic F4/80[hi] macrophages obtained from the distal part were quickly lost within 7 d (Fig 5A, left panel) because this particular colon tract is severely affected by the sulfated polysaccharides (Laroui et al, 2012). Furthermore, concomitant to the DSS-induced loss of macrophages, a massive monocyte infiltration was observed to efficiently replenish the "emptied" niche (Fig 5A, right panel), thereby restoring tissue homeostasis. As DSS was confirmed to deplete the resident macrophage niche, we treated TAM-injected GF and SPF *Kit*[MerCreMer]/*R26*[YFP] mice with DSS for 7 d and collected the distal part of the colon 3 wk later (Fig 5B). The frequency of

points showing absolute numbers of distinct myeloid cell populations present in the large intestine of SPF (n = 11) and GF (n = 8) mice. Data represent the mean ± SD. *P < 0.05; **P < 0.01, ***P < 0.001; ****P < 0.0001; ns, no significant difference; two-way ANOVA. Data represent three independent experiments. **(D)** UMAP projection of myeloid cell landscape in the colon and small intestine of SPF and corresponding GF mice. The related down-sampled events (~40,000 cells) from the colon and small intestine (SPF and GF groups) flow cytometry dataset were concatenated and exported into a single .fcs file, and the two-dimensional UMAP was generated using the concatenated sample with default parameters; DC fractions were excluded from the analysis. **(E)** Pie charts showing the proportions of F4/80[hi]Tim-4[+] and F4/80[hi]Tim-4[−] macrophages in the small and large intestines of GF (n = 8) and SPF (n = 8) mice. Data represent two independent experiments.

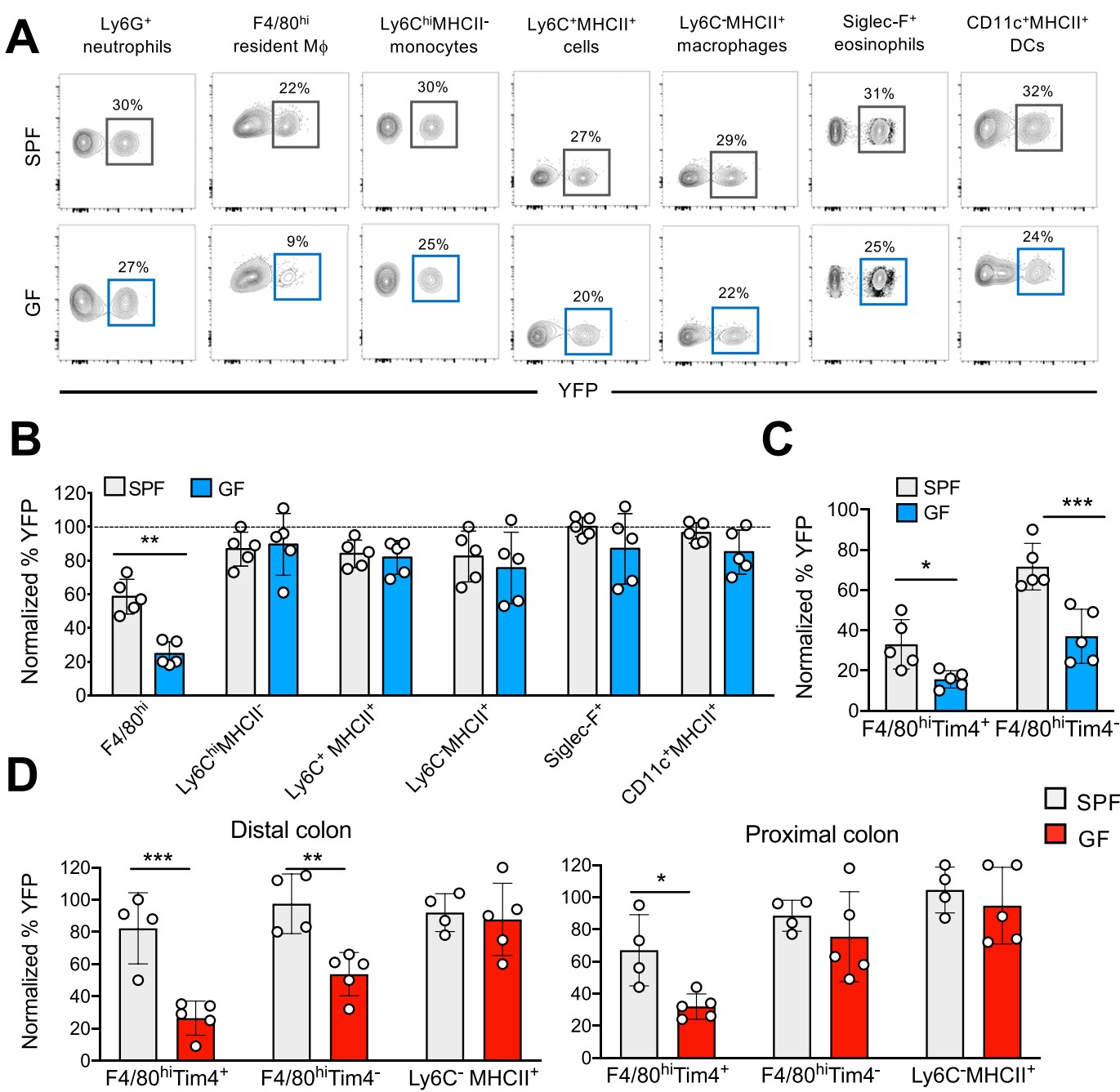

**Figure 2. Colon F4/80ʰⁱ resident macrophages from germ-free (GF) mice show slower kinetics of replacement by BM-derived cells than their specific pathogen-free (SPF) counterparts.**

**(A)** SPF and GF *Kit*^MerCreMer^/*R26*^YFP^ mice (aged 6 wk) received 4 mg tamoxifen (TAM) by oral gavage for five consecutive days and were euthanized 2 mo later. Representative flow cytometry contour plots indicating the YFP labeling of different myeloid cell populations obtained from the colon of SPF and GF mice. **(B)** Bar charts with individual data points showing the percentage of YFP⁺ colonic myeloid cells obtained from SPF and GF mice after normalization to the percentage of YFP⁺ neutrophils. Data represent the mean ± SD (n = 5 mice). **P < 0.01; two-tailed *t* test. Data represent two independent experiments. **(C)** Percentage of YFP⁺ F4/80ʰⁱTim-4⁺ and F4/80ʰⁱTim4⁻ resident macrophages obtained from SPF and GF large intestines. Percentage normalization as described in panel (B). Data represent the mean ± SD (n = 5 mice). *P < 0.05; ***P < 0.001; two-tailed *t* test. Data represent two independent experiments. **(D)** Percentage of YFP⁺ F4/80ʰⁱTim-4⁺, F4/80ʰⁱTim4⁻, and Ly6C⁻MHCII⁺monocyte-derived macrophages obtained from distal and proximal colon sections of SPF and GF mice euthanized 4 mo post-TAM treatment. Percentage normalization as described in panel (B). Data represent the mean ± SD (n = 4–5 mice). *P < 0.05; **P < 0.01; ***P < 0.001; ns, no significant difference; two-tailed *t* test. Data represent two independent experiments.

infiltrating YFP⁺ BM-derived cells into the large intestine was analyzed by flow cytometry. Under SPF conditions, both F4/80ʰⁱTim4⁺ and F4/80ʰⁱTim4⁻ macrophage subpopulations were fully replaced by YFP⁺ cells within 3 wk. Furthermore, GF mice showed significantly decreased replenishment kinetics, indicating that the refilling rate was dependent on the presence of commensals (Fig 5C and D).

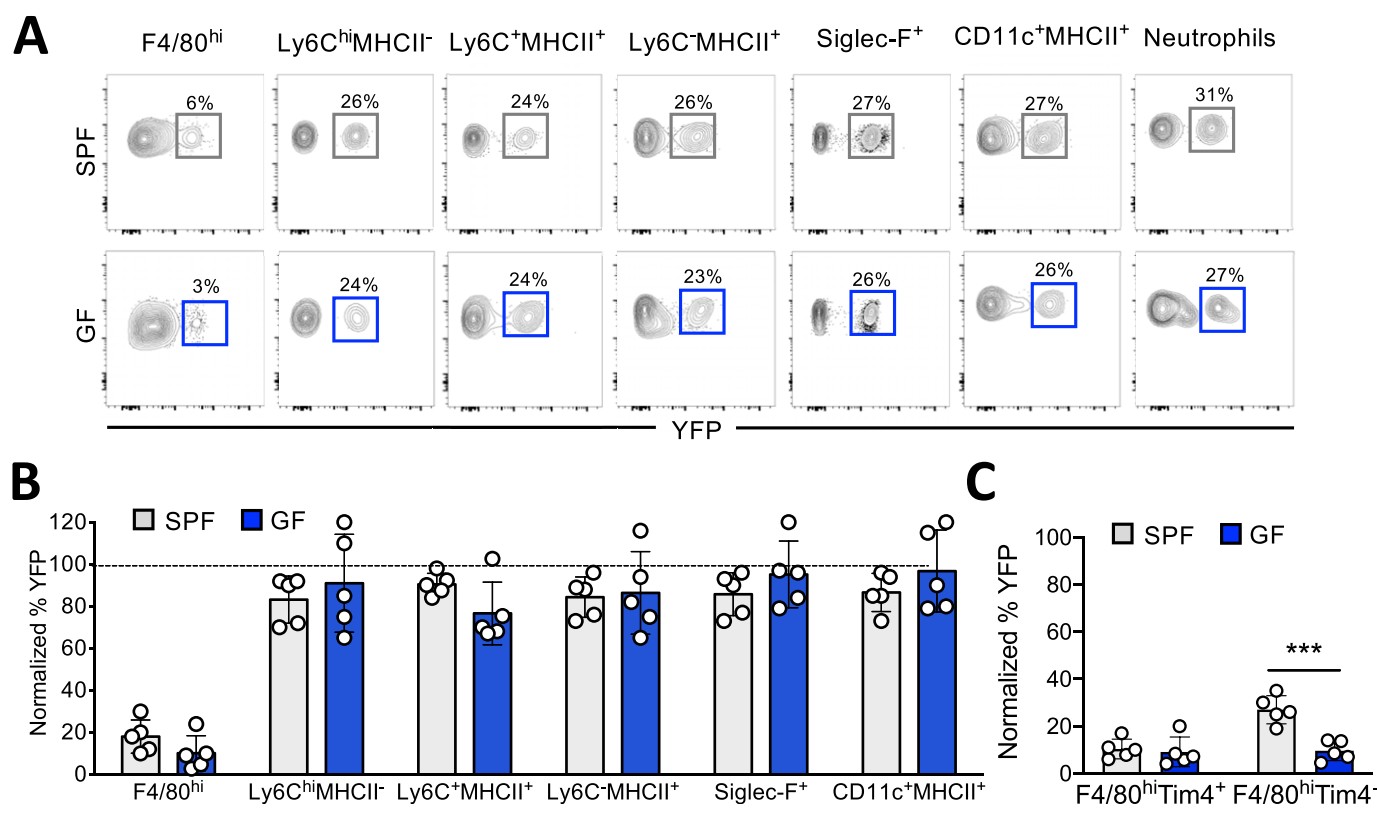

**Figure 3. F4/80^hi resident macrophages in the small intestine show minimal turnover.**
**(A)** Specific pathogen-free (SPF) and germ-free (GF) $Kit^{MerCreMer}/R26^{YFP}$ mice (aged 6 wk) received 4 mg tamoxifen (TAM) by oral gavage for five consecutive days and were euthanized 2 mo later. Representative contour plot showing the percentage of YFP⁺ F4/80^hi, Ly6C^hiMHCII⁻, Ly6C⁺MHCII⁺, Ly6C⁻MHCII⁺, Siglec-F⁺, CD11c⁺MHCII⁺, and neutrophils obtained from GF and SPF large intestines. **(B)** Bar charts with individual data points showing the percentage of YFP⁺ myeloid cells after normalization to the percentage of YFP⁺ neutrophils. Data represent the mean ± SD (n = 5 mice). Data represent two independent experiments. **(C)** Percentage of normalized YFP⁺ F4/80^hiTim-4⁺ and F4/80^hiTim4⁻ resident macrophages obtained from SPF and GF proximal small intestines. Data represent the mean ± SD (n = 5 mice). ***$P < 0.001$; two-tailed $t$ test. Data represent two independent experiments.

Based on these findings, it can be concluded that the effect of intestinal commensals on macrophage turnover is not only limited to normal gut homeostasis but also impacts the refilling kinetics of an empty available niche caused by tissue inflammation.

## Discussion

Recent advances in our understanding of macrophage ontogeny and differentiation have yielded new insights into the origin of tissue-resident innate cells (Ginhoux et al, 2010; Sheng et al, 2015; Ginhoux & Guilliams, 2016). Macrophage ontogeny can markedly differ in distinct tissue/organs because of environmental or mechanical and stress factors (Mowat et al, 2017; Dick et al, 2019). In particular, the contribution of the microbiota was proposed as one of the key components influencing the tightly regulated monocyte-dependent replenishment of macrophages resident in the intestine. In this heavily microbial-exposed organ, the local fetal-derived LP macrophages are almost completely replenished by circulating monocytes within 3 mo post-partum (Bain et al, 2014; Liu et al, 2019).

To investigate the contribution of the microbiome to macrophage replenishment kinetics, we established a GF fate-mapping

$Kit^{MerCreMer}/R26^{YFP}$ mouse strain that helped us overcome an eventual incomplete endogenous microbiome depletion through antibiotic treatment (Kennedy et al, 2018). At steady-state, the numbers of intestinal neutrophils, monocytes, and monocyte-derived macrophages were significantly diminished in the absence of endogenous microflora, which is consistent with previous reports (Bain et al, 2014). However, in contrast to previous studies (Shaw et al, 2018), we observed that the absence of commensals reduced only the absolute numbers of short-lived Tim4⁻ resident macrophages both in the large and small intestines. Furthermore, we demonstrated that the density of the microbial gut ecosystem influences the proportions of short-lived Tim4⁻ and long-lived Tim4⁺ macrophages. The small intestine, with the lowest amount of endogenous microflora, harbors the highest numbers of long-lived Tim-4⁺ resident macrophages, whereas the large intestine, with the greatest commensal load, favors dominance by the shorter-lived Tim4⁻ resident macrophage fraction.

Under homeostatic conditions, the turnover kinetics of resident macrophages was also strongly influenced by the gut microbiome. Whereas all other intestinal myeloid cells, including the monocyte-derived macrophages, were swiftly replenished by BM-derived cells independently of the endogenous microflora, commensals only selectively influenced the monocyte-replenishment of resident LP

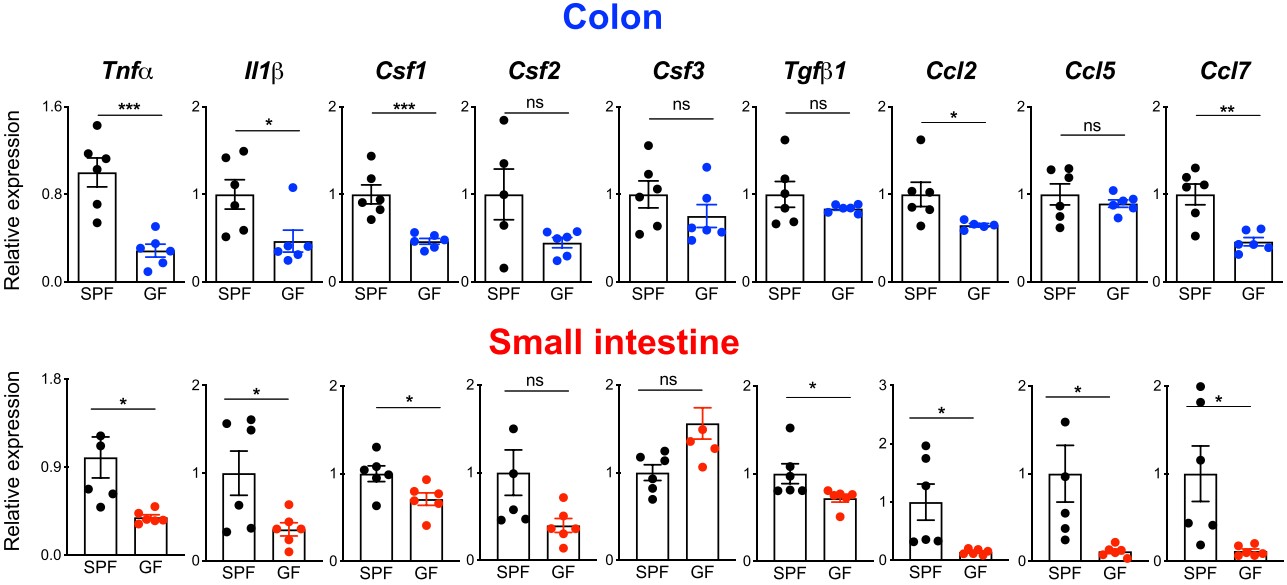

**Figure 4. The microbiota affects chemokine and cytokine expression in the intestine.**
Quantitative PCR analysis of distinct inflammatory cytokines, growth factors and chemokines in the colon (upper panels) and small intestine (lower panels) collected from germ-free and specific pathogen-free mice. Data represent the mean ± SD (n = 6 mice). *P < 0.05; **P < 0.01; ***P < 0.001; two-tailed t test.

F4/80^hi macrophages. In the GF large intestine, the turnover of both the Tim-4⁺ and Tim4⁻ macrophage subpopulations was reduced, although the long-lived Tim-4⁺ fraction showed more evident microbiota-dependency than the Tim-4⁻ fraction. On the other hand, in the small intestine, where the replenishment kinetics of the long-lived Tim-4⁺ macrophages was already extremely slow, the turnover rate of only the short-lived Tim-4⁻ macrophages was significantly affected by the endogenous microbial flora. Notably, using the DSS-treated GF *Kit*^MerCreMer^/*R26*^YFP^ mouse model, we showed that the effect of commensals is not only limited to a homeostatic macrophage replenishment but also affects the refilling efficiency of available macrophage niches caused by gut inflammation.

How the enteric microbial community influences the replenishment speed is an important question that remains to be answered and we can only speculate on possible reasons. Although the mucosal immune system tolerates endogenous commensals, they still cause a local "physiological" inflammation without exacerbating any pathogenic symptoms. Indeed, higher levels of TNF-α and IL-1β are measured in the SPF colon compared to their GF counterparts, indicating that inflammatory cytokines could be responsible for the attraction of blood circulating neutrophils and monocytes into the gut (Shi & Pamer, 2011). Indeed our analysis provides convincing evidence of lower numbers of monocytes and neutrophils in the large and small intestinal tracts. The reduced monocyte numbers could account for the slower replenishment kinetics of the local resident gut F4/80^hi macrophages but does not explain why the monocyte-derived macrophages are not affected and swiftly replaced. Other cytokines and growth factors produced locally in the intestinal microenvironment may also contribute to the monocyte differentiation process (Gross-Vered et al, 2020). Among them, the growth factor CSF1 supports differentiation of monocytes into resident tissue macrophages (Sordet et al, 2002) as

well as macrophage and DC survival (Tushinski et al, 1982; Hume et al, 1988; Bogunovic et al, 2009; Tagliani et al, 2011) in their tissue-specific niches (Guilliams et al, 2020), one of which is the gastrointestinal tract. Therefore, it can be speculated that the reduced *Csf1* levels detected in GF colons impair monocyte differentiation and macrophage survival and explain the lower numbers of F4/80^hi macrophages and their refilling kinetics observed in our study. In addition, our findings indicate that other microbial-dependent mediators, such as chemokines (Ccl2, Ccl5, and Ccl7), expressed in the intestine microenvironment are involved in the attraction of monocytes and, therefore, in controlling the turnover rates of resident macrophages.

In summary, the microbiota favors the monocyte-dependent replenishment kinetics of resident LP intestinal F4/80^hi macrophages, possibly, among other factors, by modulating the expression of specific cytokines and chemokines in the intestinal microenvironment. Moreover, during gastrointestinal inflammation, commensals support rapid replenishment of the perturbed macrophage "niche," promoting a rapid restoration of functional gut homeostasis.

## Materials and Methods

### Mice

Fate-mapping *Kit*^MerCreMer^/*R26*^YFP^ mice were generated as previously described (Sheng et al, 2015) and bred and maintained in a SPF animal facility at the Nanyang Technological University. The corresponding GF *Kit*^MerCreMer^/*R26*^YFP^ mouse line was generated by the Nanyang Technological University GF mouse facility and housed in flexible-film isolators supplied with HEPA-filtered air.

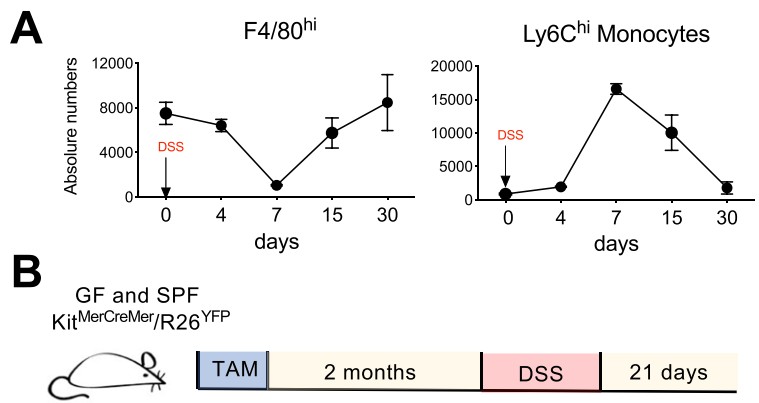

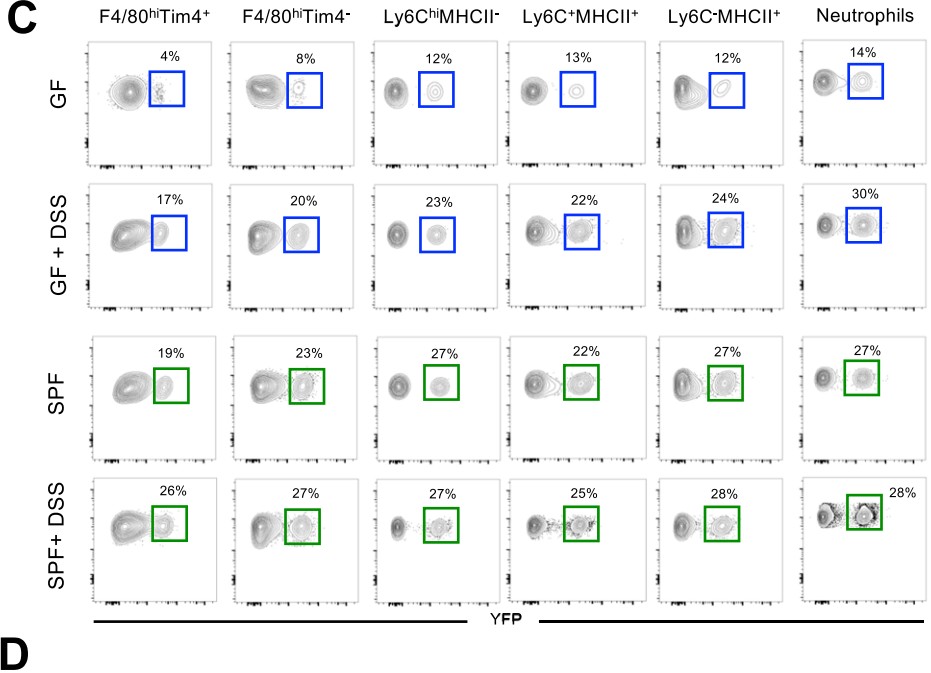

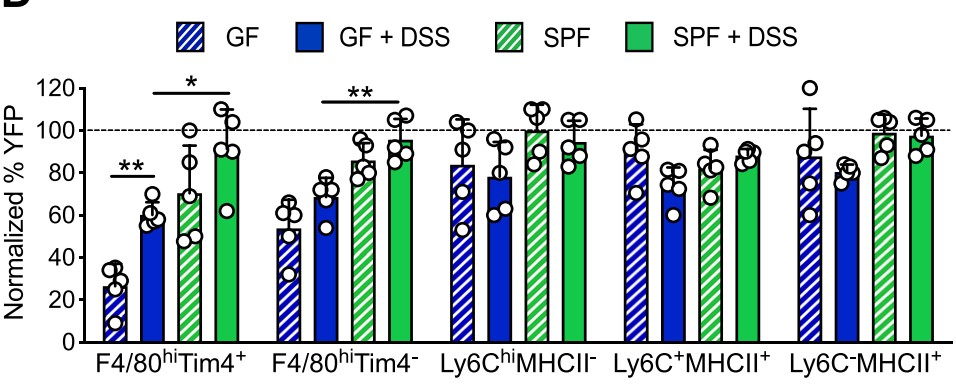

**Figure 5. Germ-free (GF) mice show ineffective niche refilling.**
**(A)** Profile analysis of colon lamina propria F4/80[hi] resident macrophages and Ly6C[hi] monocytes during dextran sulfate sodium (DSS) treatment (7 d) and the post-recovery period (15 and 30 d) in specific pathogen-free (SPF) WT C57/BL6 mice. Absolute numbers are shown. Data represent the mean ± SD (3–5 mice for each time point). **(B)** Schematic diagram of the experimental procedure used for the fate-mapping mouse line. **(C)** TAM-treated SPF and GF *Kit*[MerCreMer]/*R26*[YFP] mice were euthanized and analyzed 21 d post-DSS treatment. Representative contour plots showing the percentage of YFP⁺ F4/80[hi]Tim-4⁺, F4/80[hi]Tim-4⁻, Ly6C[hi]MHCII⁻, Ly6C⁺MHCII⁺, Ly6C⁻MHCII⁺, and neutrophils obtained from GF (steady-state and DSS-treated, upper panels) and SPF (steady-state and DSS-treated, lower panels) from distal large intestines. **(D)** Bar charts with individual data points showing the percentage of normalized YFP⁺ F4/80[hi]Tim-4⁺, F4/80[hi]Tim-4⁻, Ly6C[hi]MHCII⁻, Ly6C⁺MHCII⁺, and Ly6C⁻MHCII⁺ obtained from SPF (steady-state and DSS-treated) and GF (steady-state and DSS-treated) distal large intestines. Data represent the mean ± SD (n = 5 mice/group).
*$P < 0.05$; **$P < 0.01$; two-way ANOVA. Data represent four independent experiments.

The mice were given autoclaved food and sterile tap water. GF status was verified through the collection of fecal samples, which were routinely tested for microbes and parasites. The mice used in this study were confirmed to be free of all bacteria, fungi, and metazoans.

All studies involving mice were carried out according to the recommendations of the National Advisory Committee for Laboratory Animal Research and ARF SBS/NIE 18044, 18081, and 19093 protocols were approved by the Institutional Animal Care and Use Committee of the Nanyang Technological University.

### Tamoxifen-inducible adult fate-mapping mouse model

$Kit^{MerCreMer}/R26^{YFP}$ female fate-mapping mice were used to determine the turnover kinetics of distinct myeloid cell subsets under SPF and GF conditions. For adult labeling, 6- to 8-wk-old mice received 4 mg tamoxifen (TAM) (Sigma-Aldrich) by oral gavage for five consecutive days as previously described (Sheng et al, 2015). Upon TAM injection, the YFP label will be induced in all c-kit⁺ cells, predominantly residing in the BM, and these cells will retain the YFP label once they left the BM and seeded into the periphery, including the gut. Mice were euthanized at different time-points (2, 3, and 4 mo) to monitor the progression of macrophage replenishment. Colon and small intestine tissues were collected for subsequent cell isolation, and multiparameter flow cytometry cell analysis. The YFP labeling levels between different cell types were normalized to that of short-lived neutrophils, to minimize the unavoidable variation in absolute marking after tamoxifen injections between mice (Sheng et al, 2015).

### Experimental acute colitis model-dextran sodium sulfate treatment

TAM-gavaged GF and SPF $Kit^{MerCreMer}/R26^{YFP}$ fate-mapping mice were treated with 2% DSS (50,000 D; MP Biomedical) supplied in the drinking water for seven consecutive days. This DSS concentration causes mild colon inflammation and no lethality of treated mice (Soncin et al, 2018). After 1 wk, the DSS was replaced with drinking water and animals analyzed 3 wk later.

### Intestinal macrophage isolation

Intestinal macrophages were isolated as previously described (Muzaki et al, 2016). Briefly, the colon and small intestine were collected from SPF and GF mice, opened, and rinsed with phosphate-buffered saline to remove the luminal contents. To remove the epithelium, the colon and small intestine were incubated in 25 ml Hanks' balanced salt solution without Ca⁺⁺ and Mg⁺⁺ with 1.3 mM EDTA under shaking conditions at 37°C (1 h for colon and 35 min for small intestine). After incubation, the colon and small intestine were washed in 2% IMDM and then cut into small pieces, which subsequently were digested in 2% IMDM containing 1 mg/ml collagenase D (Roche) and 20 U/ml DNase I (Life Technologies) under shaking conditions at 37°C (75 min for colon and 60 min for small intestine). The digested tissue was then gently mashed through a 150-$\mu$m cell strainer. The leukocyte population was enriched using a 70%/40% Percoll gradient (GE Healthcare Life Science). Low-density cells at the interface were harvested and processed further for flow cytometric analysis.

### Flow cytometry staining

Single-cell suspensions were stained and subsequently analyzed using a BD LSRFortessa or FACSymphony A3 five laser flow cytometer (BD Bioscience). Data were analyzed using FlowJo software (TreeStar). The following antibodies were used: CD11c (clone: N418), CD11b (clone: M1/70), Ly6C (HK1.4), Ly6G (clone: 1A8), Tim-4 (clone:RMT4-54), and EMR1 (also known as F4/80) (clone: BM8) were purchased from BioLegend. CD45 (clone: 30-F11) and Siglec-F (clone E50-2440) were purchased from BD Biosciences. MHCII (clone:M5/114.15.2) and CD103 (clone 2E7) were purchased from Thermo Fisher Scientific.

### Quantitative PCR analysis

cDNA was generated using SuperScript III Reverse Transcriptase (Invitrogen) according to the manufacturer's instructions. Quantitative real-time PCR (qRT-PCR) was then performed using the SYBR green chemistry method (KabaBiosystem). Reactions were run on a real-time qPCR system (Illumina). Relative mRNA expression was normalized to $\beta$-actin, and data were presented as the median fold-change of triplicate samples compared with WT controls unless otherwise stated. The primer sequences were as follows: $Csf1$; Fwd: ggtggaactgccagtatagaaag, Rev: tcccatatgtctccttccataaa; $\beta$-$Actin$; Fwd: aaggccaaccgtgaaaagat, Rev: cctgtggtacgaccagaggcataca; $Csf2$; Fwd: gcatgtagaggccatcaaaga, Rev: cgggtctgcacacatgtta; $Csf3$; Fwd: gctgctggagcagttgtg, Rev: gggatccccagagagtgg; $Tnf\alpha$; Fwd: tggagcaacatgtggaactc, Rev: gtcagcagccggttacca; $Il1\beta$; Fwd: gggcctcaaaggaaagaatc, Rev: ttcttcttttgggtattgcttgg; $Ccl2$; Fwd: ttaaaaacctggatcggaaccaa, Rev: gcattagcttcagatttacgggt; $TGF\beta1$; Fwd: tggagcaacatgtggaactc, Rev: gtcagcagccggttacca; $Ccl5$; Fwd: gcagcaagtgctccaatctt, Rev: acttcttctctgggttggca; $Ccl7$; Fwd: ttctgtgcctgctgctcata, Rev: cttctgtagctcttgagattcctct.

### Statistical analysis

Statistical analysis was performed using GraphPad Prism 9.0.1 software (GraphPad Software). All values were expressed as the mean ± standard deviation of the mean (SD) as indicated in the legend. Samples were analyzed by $t$ test (two-tailed) or Bonferroni two-way ANOVA. A $P$-value < 0.05 was considered to indicate statistical significance.

# Data Availability

The original flow cytometry data have been deposited in the NTU Open Access Data Repository (DR-NTU) https://doi.org/10.21979/N9/XBXJPP.

# Supplementary Information

# Acknowledgements

The authors would like to thank Norhashimah Binte Sulaiman for excellent germ-free mouse management and Insight Editing London for proofreading the manuscript before submission. This work was supported by a Ministry of Education Tier1 grant awarded to C Ruedl.

## Author Contributions

Q Chen: data curation, formal analysis, investigation, visualization, and methodology.

S Nair: investigation and methodology.

C Ruedl: conceptualization, resources, data curation, formal analysis, supervision, funding acquisition, validation, project administration, and writing—original draft, review, and editing.

## Conflict of Interest Statement

The authors declare that they have no conflict of interest.

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
