## [Reviewer comments · Life Science Alliance]

Life Science Alliance

Microbiota regulates the turnover kinetics of gut macrophages in health and inflammation

Christiane Ruedl, Qi Chen, and Sajith Kumar Nair

DOI: <https://doi.org/10.26508/lsa.202101178>

Corresponding author(s): *Christiane Ruedl, Nanyang Technological University*

Review Timeline:

Submission Date:	2021-07-28
Editorial Decision:	2021-09-20
Revision Received:	2021-09-28
Editorial Decision:	2021-10-13
Revision Received:	2021-10-19
Accepted:	2021-10-20

Scientific Editor: Novella Guidi

Transaction Report:

September 20, 2021

Re: Life Science Alliance manuscript #LSA-2021-01178

Prof. Christiane Ruedl
Nanyang Technological University
School of Biological Sciences
60 Nanyang Drive
Singapore 637551

Dear Dr. Ruedl,

Thank you for submitting your manuscript entitled "Microbiota regulates the turnover kinetics of intestinal macrophages" to Life Science Alliance. The manuscript was assessed by an expert reviewer, whose comments are appended to this letter. We invite you to submit a revised manuscript addressing the Reviewer's comments.

The typical timeframe for revisions is three months. Please note that papers are generally considered through only one revision cycle, so strong support from the referee on the revised version is needed for acceptance.

When submitting the revision, please include a letter addressing the reviewer's comments point by point.

Thank you for this interesting contribution to Life Science Alliance. We are looking forward to receiving your revised manuscript.

Sincerely,

B. MANUSCRIPT ORGANIZATION AND FORMATTING:

Reviewer #1 (Comments to the Authors (Required)):

In the current study, authors demonstrated that microbiota affects F4/80(hi) resident macrophage turnover in the intestine. Authors utilized fate-mapping mouse system which labels BM originated cells and showed that various myeloid cells are replenished by BM. To assess the cell type-specific contribution of microbiota during their turnover, mice were also housed in germ-free condition. Colonic F4/80(hi) macrophages were the most affected cell type by microbiota in terms of cell turnover under the steady-state and inflammatory settings, suggesting that microbial stimulation supports F4/80(hi) macrophage turnover. In small intestine where microbes are less populated, F4/80(hi) macrophage turnover was less prominent, especially for long-lived Tim-4(+) subpopulation. Although the study provides the evidence of microbial involvement in F4/80(hi) macrophage replenishment, its detailed mechanism is still obscure. Still this approach provides useful information to the researchers in related fields.

Specific comments:

1. F4/80(hi)MHCII(+) in Figure 1C seems monocyte, which should be F4/80(hi)MHCII(-).
2. Did authors checked YFP expression of c-kit(+) cells in the BM at 2~4 months after TAM treatment? It would be important to check whether YFP(-)c-kit(+) cells are not present in TAM-treated mice at the time point when analyzed. Also, please explain why authors used different time points after TAM treatment.
3. Figures need to be labelled following order they appear in the text. Currently Figure 2E appears after Figure 3B. Alternatively, Figure 2E and 3B can be combined and presented as new set of figure.
4. Please show representative plots, at least for F4/80(hi) macrophages in Figure 3. Also, representative plot for Figure 4C would be informative.
5. Infection and injury induces influx of various cell types but number of resident macrophages are generally less affected. DSS obviously increases cell influx to large intestine and relative ratio of F4/80(hi) macrophage should be decreased. Authors need to show absolute number of F4/80(hi) cells in Figure 4A to show this population actually "lost".

We thank all the reviewer for his/her constructive comments on our work. We have tried to address as well as we can on the experimental concerns and provide detailed answers to the raised remarks. We have included a new Fig. 4 and revised the old ones based on the comments. Amendments in the manuscript have been highlighted in red.

We hope that with the described changes and responses, the paper is now suitable for publication in **LSA**.

Reviewer #1 (Comments to the Authors (Required)):

In the current study, authors demonstrated that microbiota affects F4/80(hi) resident macrophage turnover in the intestine. Authors utilized fate-mapping mouse system which labels BM originated cells and showed that various myeloid cells are replenished by BM. To assess the cell type-specific contribution of microbiota during their turnover, mice were also housed in germ-free condition. Colonic F4/80(hi) macrophages were the most affected cell type by microbiota in terms of cell turnover under the steady-state and inflammatory settings, suggesting that microbial stimulation supports F4/80(hi) macrophage turnover. In small intestine where microbes are less populated, F4/80(hi) macrophage turnover was less prominent, especially for long-lived Tim-4(+) subpopulation. Although the study provides the evidence of microbial involvement in F4/80(hi) macrophage replenishment, its detailed mechanism is still obscure. Still this approach provides useful information to the researchers in related fields.

Specific comments:

1. F4/80^{hi}MHCII⁺ in Figure 1C seems monocyte, which should be F4/80(hi)MHCII(-).

I guess the reviewer meant Ly6C^{hi}MHCII⁺ ? We have corrected this typo Ly6C^{hi}MHCII⁺ into Ly6C^{hi}MHCII⁻ in Fig. 1C and Suppl. Figure 1.

2. Did authors checked YFP expression of c-kit(+) cells in the BM at 2~4 months after TAM treatment? It would be important to check whether YFP(-)c-kit(+) cells are not present in TAM-treated mice at the time point when analyzed. Also, please explain why authors used different time points after TAM treatment.

Since HSC are expressing c-kit (also known as stem cell factor receptor), a 5 day treatment with TAM will maintain the YFP signal in the BM over time, hence there will be a continuous seeding of YFP⁺ BM cells into the periphery. Depending on their turnover rates distinct lymphoid and myeloid cells will be labelled at different kinetics.

In particular, some specific intestinal macrophage subpopulations (e.g. Tim-4⁺) are known to be replaced slowly over time instead some other subpopulations are swiftly refilled (Tim-4⁻). Therefore to capture more efficiently the refilling kinetics at steady-state, we have chosen two different time-points (2 [Fig. 2 B, C and Fig. 3] and 4 months [Fig. 2D]). On the other hand, under inflammatory conditions (e.g. DSS treatment) we opted for a shorter window (1 month) since we expected a faster monocyte-mediated replenishment process.

3. Figures need to be labelled following order they appear in the text. Currently Figure 2E appears after Figure 3B. Alternatively, Figure 2E and 3B can be combined and presented as new set of figure.

As suggested, we have included a new figure 4 which included the cytokine and chemokine qPCR analysis of colon and small intestine.

4. Please show representative plots, at least for F4/80(hi) macrophages in Figure 3. Also, representative plot for Figure 4C would be informative.

As suggested, representative flow cytometry plots were included in both Fig. 3 and new Fig. 5C.

5. Infection and injury induces influx of various cell types but number of resident macrophages are generally less affected. DSS obviously increases cell influx to large intestine and relative ratio of F4/80(hi) macrophage should be decreased. Authors need to show absolute number of F4/80(hi) cells in Figure 4A to show this population actually "lost".

We agree with the reviewer that sometimes the frequency does not reflect a reduction or increase in particular if there is a massive infiltration of another cell population. We have added changed the % in absolute numbers (New Fig. 5A).

October 13, 2021

RE: Life Science Alliance Manuscript #LSA-2021-01178R

Prof. Christiane Ruedl
Nanyang Technological University
School of Biological Sciences
60 Nanyang Drive
Singapore 637551
Singapore

Dear Dr. Ruedl,

Thank you for submitting your revised manuscript entitled "Microbiota regulates the turnover kinetics of intestinal macrophages". We would be happy to publish your paper in Life Science Alliance pending final revisions necessary to meet our formatting guidelines.

- As a one of main myeloid cell population, Ly6C+MHCII+ cells need to be included in Figures 2,3,5. Also, please update text accordingly
- please add the Twitter handle of your host institute/organization as well as your own or/and one of the authors in our system
- please add Abstract in our system
- please note that titles in the system and the manuscript file must match
- please consult our manuscript preparation guidelines <https://www.life-science-alliance.org/manuscript-prep> and make sure your manuscript sections are in the correct order and labeled correctly
- please use the [10 author names, et al.] format in your references (i.e. limit the author names to the first 10)
- please label Figure 5 correctly
- please provide the link to access to original flow cytometry data in your data availability section

A. FINAL FILES:

B. MANUSCRIPT ORGANIZATION AND FORMATTING:

Sincerely,

Reviewer #2 (Comments to the Authors (Required)):

Authors addressed comments raised by this reviewer.

Minor point:

As a one of main myeloid cell population, Ly6C+MHCII+ cells need to be included in Figures 2,3,5. Also, please update text accordingly.

October 20, 2021

RE: Life Science Alliance Manuscript #LSA-2021-01178RR

Prof. Christiane Ruedl
Nanyang Technological University
School of Biological Sciences
60 Nanyang Drive
Singapore 637551
Singapore

Dear Dr. Ruedl,

Thank you for submitting your Research Article entitled "Microbiota regulates the turnover kinetics of gut macrophages in health and inflammation". It is a pleasure to let you know that your manuscript is now accepted for publication in Life Science Alliance. Congratulations on this interesting work.

DISTRIBUTION OF MATERIALS:

Again, congratulations on a very nice paper. I hope you found the review process to be constructive and are pleased with how the manuscript was handled editorially. We look forward to future exciting submissions from your lab.

Sincerely,
